# Testing the Effectiveness of the Diagnostic Probing Paradigm on Italian Treebanks

**Alessio Miaschi** *[ID]**, Chiara Alzetta** [ID]**, Dominique Brunato** [ID]**, Felice Dell'Orletta** *[ID] **and Giulia Venturi** [ID]

CNR—Institute for Computational Linguistics "A. Zampolli", ItaliaNLPLab, Via G. Moruzzi 1, 56124 Pisa, Italy
* Correspondence: alessio.miaschi@ilc.cnr.it (A.M.); felice.dellorletta@ilc.cnr.it (F.D.)

**Abstract:** The outstanding performance recently reached by neural language models (NLMs) across many natural language processing (NLP) tasks has steered the debate towards understanding whether NLMs implicitly learn linguistic competence. Probes, i.e., supervised models trained using NLM representations to predict linguistic properties, are frequently adopted to investigate this issue. However, it is still questioned if probing classification tasks really enable such investigation or if they simply hint at surface patterns in the data. This work contributes to this debate by presenting an approach to assessing the effectiveness of a suite of probing tasks aimed at testing the linguistic knowledge implicitly encoded by one of the most prominent NLMs, BERT. To this aim, we compared the performance of probes when predicting gold and automatically altered values of a set of linguistic features. Our experiments were performed on Italian and were evaluated across BERT's layers and for sentences with different lengths. As a general result, we observed higher performance in the prediction of gold values, thus suggesting that the probing model is sensitive to the distortion of feature values. However, our experiments also showed that the length of a sentence is a highly influential factor that is able to confound the probing model's predictions.

**Keywords:** neural language models; BERT; probing tasks; treebanks; Italian language





## 1. Introduction

The rise of large pre-trained neural language models (NLMs) has revolutionized the field of natural language processing (NLP) in the last five years. In particular, the introduction of deep contextualized models based on the Transformer architecture [1], which is able to learn word vectors that are sensitive to the context in which words appear, has yielded significant improvements in many NLP tasks [2–4]. Even with some differences concerning the sizes of their parameters, architectures, and training datasets [5–7], these models are all pre-trained on large amounts of text and, subsequently, fine-tuned on task-specific, supervised downstream tasks. Among the many Transformer-based models, BERT (Bidirectional Encoder Representations from Transformers) was the first one to push the state of the art in many areas of NLP [8].

However, it is well known in the literature that the remarkable ability of BERT—and of NLMs in general—to perform numerous language-understanding tasks goes with an opaqueness concerning the interpretation of their internal mechanisms. Particular interest has been devoted in the last few years to the investigation of the linguistic abilities implicitly encoded by models [9]. Namely, several methods have been proposed to obtain meaningful explanations of how NLMs are able to capture syntax- and semantic-sensitive phenomena [10], also taking inspiration from human language experiments [11,12]. They range from the analysis of attention mechanisms [13] and the definition of diagnostic tests [14] to the implementation of explainability techniques via, e.g., integrated gradients [15]. One of the most explored methods is the definition of probing tasks, which a model can solve only if it has encoded a precise linguistic phenomenon within its representations [16].

However, despite the amount of work based on the diagnostic probing approach, as outlined by Belinkov [17], there are still several open questions, such as the following:

Which probing model should we use to assess the linguistic competence of an NLM? Are probes the most effective strategy for achieving such a goal? These questions fostered two complementary lines of research. The first one is devoted to modifying the architectures of current probing models; the other one is focused on evaluating their effectiveness. Both are still not well-investigated issues, although their importance for advancing the research on the evaluation of NLMs' linguistic competencies has been widely recognized.

This study would contribute to the debate on the effectiveness of the probing paradigm as a diagnostic method to assess the linguistic knowledge implicitly encoded by BERT. To achieve this goal, we defined a multifaced approach that comprised a number of experiments aimed at comparing the performance of a probing model that was trained using BERT representations to predict the values of a set of sentence-level properties extracted from the Italian Universal Dependency Treebank [18] and from a suite of control datasets that we specifically built for the purpose of this study. Starting with and extending the methodology introduced by Miaschi et al. [19], we define as the control dataset a set of linguistic features whose values are automatically altered in order to be increasingly different from the values in the treebank, which are referred to as *gold* values. Our underlying hypothesis is the following: If the probing model's predictions of the variously altered values diverge from the predictions of the gold values, this possibly suggests that the corresponding probing tasks are effective strategies for testing the linguistic knowledge embedded in BERT representations. We will discuss the results of the experiments in light of this hypothesis. The remainder of this paper is organized as follows. We present our background and related work in Section 2. Section 3 introduces our methodology and presents the data, the monitored linguistic features, and the models used in the study. Section 4 presents the results, and in Section 5, we will draw the conclusions.

*Contributions*

With respect to the previous literature, the main contributions of our work lie in the following points:

- We present a methodology for testing the reliability of probing tasks by building control datasets at diverse levels of complexity;
- We assess the extent to which the linguistic knowledge encoded by BERT is influenced by the length of a sentence and how the length can represent a confounding factor that may bias the real estimate of BERT's knowledge of a wide variety of (morpho-)syntactic phenomena;
- We test the effectiveness of the diagnostic probing task approach on Italian, a language frequently neglected by studies on probing.

## 2. The Diagnostic Probing Paradigm

In the last few years, the analysis of the inner workings of state-of-the-art neural language models (NLMs) has become one of the most popular lines of research in NLP. In particular, great efforts have been devoted to obtaining meaningful explanations about their linguistic competence in order to understand the extent to which these models are able to capture linguistic properties targeting a variety of domains [20]. These approaches range from the definition of fill-the-gap probes [14] and probing tasks that a model can only solve if it has encoded a precise linguistic phenomenon [16,21,22] to the analysis of attention mechanisms [23–25] and correlations between representations [26].

Among the different strategies developed to study the implicit language competencies encoded by NLMs, the *diagnostic probing task* approach has emerged as one of the most commonly adopted ones. The idea behind the probing paradigm is actually quite simple: using a diagnostic classifier, the *probing model* or *probe*, which takes the output representations of an NLM as input, to perform a *probing task*, e.g., to predict a given language property. If the probing model correctly predicts the property, then we can assume that the representations somehow encode that property.

Studies relying on this approach reported that NLMs' contextual representations are able to encode a broad spectrum of linguistic properties, from information about parts of speech (POSs) and other morphological properties to syntactic and semantic information. In particular, these works demonstrated that NLMs learn a variety of language properties in a hierarchical manner [10,27,28] and that their representations also support the extraction of dependency parse trees [29]. By training a simple probing classifier that has access only to the per-token contextual embeddings of a BERT model, Tenney et al. [30] showed that the order in which specific abstractions are encoded within the internal representations reflects the traditional hierarchy of the NLP pipeline: POS tags are processed earliest, followed by constituents, dependencies, semantic roles, and coreference. Liu et al. [31], instead, quantified differences in the transferability of individual layers between different NLMs, showing that higher layers of ELMo [32] are more task-specific (less general), while transformer layers (BERT) do not exhibit this increase in task-specificity.

Despite this emerging body of work, there are still several open questions about how probing tasks should be designed, how complex a probe should be allowed to be, and whether probes actually show the linguistic generalization abilities of NLMs rather than learning the linguistic tasks [17]. In the first line of research, which deals with the design of probing classifiers, several works investigated which model should be used as a probe and which metric should be employed to measure the performance. In this respect, it is still questioned if one should rely on simple models [29,31,33] or more complex ones [34,35] in terms of model parametrization. For instance, Voita and Titov [35] suggested designing alternative probes by using a novel information-theoretic approach that balanced a probe's inner complexity with its task performance. Although this line of research raises many interesting questions, in this work, we take distance from it and investigate the probing paradigm from a different viewpoint.

Our perspective is closer to the second line of research on the probing task approach, which, indeed, is concerned with testing the evaluation of the effectiveness of probing models. Embracing such a line, for example, Hewitt and Liang [21] suggested that probing tasks might conceal the information about an NLM representation behind the ability of a probe to learn surface patterns in data. To test this intuition, they introduced the idea of *control tasks*, a set of tasks that associated word types with random outputs that could be solved by simply learning regularities. Measuring the difference between the accuracy on linguistic tasks and on control tasks (a property defined as *selectivity*), they identified 'good' probes as the ones for which the model achieved high linguistic task accuracy and low control task accuracy, thus providing insights into the linguistic properties of a representation. Along the same line, Ravichander et al. [36] tested probing tasks by creating control datasets in which a property was always reported in a dataset with the same value; thus, it was not discriminative for testing the information contained in the representations. Their experiments highlighted that a probe may also incidentally learn a property, thus casting doubts on the effectiveness of probing tasks.

While sharing the same goal as that in these previous works, our study differs in two main respects. Firstly, we followed an approach similar to that of Hewitt and Liang [21], but we introduce a methodology for progressively testing the effectiveness of probing models by devising diverse control tasks differing at the level of increasing complexity and intending to address a larger set of linguistic phenomena. Secondly, we focus on the Italian language, which is much less explored in the area of interpretability. In fact, the majority of research is focused on English or, at most, multilingual models, with only a few exceptions [37–39].

## 3. Methodology

The methodology that we devised is aimed at testing whether a diagnostic probing model really encodes the linguistic competencies of an NLM or simply learns the regularities of one or more probing tasks. To this aim, we trained a probing model by using BERT sentence representations, as described in Section 3.4, and then tested its performance in

the resolution of a set of linguistic tasks. These tasks consisted of predicting the values of various linguistic features (see Section 3.2) extracted from different sections of the Italian Universal Dependency Treebank (IUDT).

The probing model was tested in two main scenarios. In the first one, the model had to predict gold features, i.e., the real values of the features in IUDT sentences. In the second scenario, the gold values were altered based on multiple strategies in order to obtain alternative datasets at different control levels. As discussed in Section 3.3, this scenario, which was articulated into multiple ones, was based on the rationale that if the predictions of the probing model were more accurate and, thus, more similar to the gold values than to the automatically altered ones, then we might assume that BERT's representations do encode the linguistic knowledge required to solve the task. Consequently, the intuition is that the probing model has not simply learned some regularities that are possibly found in the dataset and used them to solve the linguistic task.

*3.1. Data*

For our experiments, we relied on the Italian Universal Dependencies Treebank (IUDT), version 2.5. The IUDT contains a total of 35,480 sentences and 811,488 tokens, and it consists of a combination of four sections that are representative of the standard Italian language, i.e., the Italian version of the multilingual Turin University Parallel Treebank (ParTUT) [40], the Venice Italian Treebank (VIT) [41], the Italian Stanford Dependency Treebank (ISDT) [42], PUD [43], and of two sections including examples of social media texts, i.e., PoSTWITA [44] and TWITTIRÒ [45].

Considering the high variability in terms of sentence length in the IUDT, which contains sentences ranging from 1 to 310 tokens long, we decided to split the treebank into three subsets, which contained the shortest, the standard, and the longest sentences. The larger subset was the Standard one; it contained 21,991 sentences with a length between 10 and 30 tokens. This is a quite typical length in Italian, a language in which the average sentence length is equal to about 20 tokens, such as in this example sentence acquired from the Standard subset: '*Un rumore infernale, simile al passaggio di un treno, risuona nei corridoi sotterranei che solcano Rochester*' (trad. 'An infernal noise, similar to the passage of a train, resounds in the underground corridors that run through Rochester').

The other two subsets comprised sentences whose lengths were less standard. Within the Shortest subset, we included 5538 sentences whose length was up to 9 tokens. This set covered many examples of nominal or elliptical sentences, including, for instance, news titles (e.g., '*Battesimo per l'opera verdiana.*', trad. 'Baptism for Verdi's opera.'), short questions (e.g., '*Come si spiega un simile risultato?*', trad. 'How can such a result be explained?'), and sentences showing a quite simple syntactic structure (e.g., '*Questa ricchezza è tutta apparenza.*', trad. 'This wealth is all appearance'). Note that, for this subset, we excluded sentences with less than 3 tokens (288 in the dataset), since they do not show a proper syntactic structure given that they generally consist of a single token plus punctuation. The set of long sentences, on the other hand, comprised sentences whose length ranged between 31 and 100 tokens, and it contained 7585 sentences. The following 58-token-long sentence represents a quite typical example of sentences belonging to the Longest subset: '*Una giornata convulsa durante la quale il presidente della Regione Lazio, Renata Polverini, è arrivata vicina alle dimissioni in seguito alla crisi generata dall'abuso di fondi pubblici da parte del Pdl laziale per il quale è indagato, con l'accusa di peculato, l'ex capogruppo Franco Fiorito.*', trad. 'It was a convulsive day during which the President of the Lazio Region, Renata Polverini, came close to resigning following the crisis generated by the misuse of public funds by the Lazio PDL, for which former group leader Franco Fiorito is under investigation on charges of embezzlement.'. The IUDT reported an additional 78 sentences longer than 100 tokens, which we excluded from the experiments since we noticed that they were characterized by a debatable annotation, possibly caused by an erroneous sentence splitting. Note that, for the specific purpose of the experiments conducted in this study, we undersampled the

Longest set to 5538 sentences, which we randomly selected, in order to balance it to the set of sentences in the Shortest subset.

*3.2. Linguistic Features*

In order to probe the linguistic competence encoded by the language model, we relied on the approach proposed for the first time by Miaschi et al. [22], which consisted of predicting the values of multiple linguistic features of a sentence by using the model's representations. The set of linguistic features was based on the one described in Brunato et al. [46], which included about 130 features representative of the linguistic structure underlying a sentence and derived from raw, morpho-syntactic, and syntactic levels of annotation. In this study, we selected the 77 most frequent features occurring in the IUDT sections in order to prevent data sparsity issues. As can be seen in Table 1, they were grouped into seven main types of linguistic phenomena, which ranged from morpho-syntactic and inflectional properties to more complex aspects of sentence structure (e.g., the depth of the whole syntactic tree), to features referring to the structures of specific sub-trees, such as the relative order of subjects and objects with respect to the verb, and to the use of subordination.

**Table 1.** Probing features used in the experiments grouped into seven main types of linguistic phenomena.

| Linguistic Feature | Label |
| --- | --- |
| **Order of elements (*Order*)** | |
| Relative order of subject and object | subj_pre, subj_post, obj_post |
| **Morpho-syntactic information (*POS*)** | |
| Distribution of UD and language-specific POS | upos_dist_*, xpos_dist_* |
| **Use of Subordination (*Subord*)** | |
| Distribution of subordinate clauses | subordinate_prop_dist |
| Average length of subordination chains and distribution by depth | avg_subord_chain_len, subordinate_dist_1 |
| Relative order of subordinate clauses | subordinate_post |
| **Syntactic Relations (*SyntacticDep*)** | |
| Distribution of dependency relations | dep_dist_* |
| **Global and Local Parsed Tree Structures (*TreeStructure*)** | |
| Depth of the whole syntactic tree | parse_depth |
| Average length of dependency links and of the longest link | avg_links_len, max_links_len |
| Average length of prepositional chains and distribution by depth | avg_prep_chain_len, prep_dist_1 |
| Clause length | avg_token_per_clause |
| **Inflectional morphology (*VerbInflection*)** | |
| Inflectional morphology of lexical verbs and auxiliaries | verbs_*, aux_* |
| **Verbal Predicate Structure (*VerbPredicate*)** | |
| Distribution of verbal heads and verbal roots | verbal_head_dist, verbal_root_perc |
| Verb arity and distribution of verbs by arity | avg_verb_edges, verbal_arity_* |

We chose to rely on these features for two main reasons. Firstly, they have been shown to be highly predictive when leveraged by traditional learning models on various classification problems where linguistic information plays a fundamental role [46]. In addition, they are multilingual, as they are based on the Universal Dependency formalism for sentence representation [47]. In fact, they were successfully used to profile the knowledge encoded in the language representations of contextual NLMs for both the Italian [38] and English languages [22].

Figure 1 exemplifies some of them that were extracted from the following sentence acquired from the Standard subset:

(1)  *In Svizzera, alcuni militanti si sono arrampicati sul tetto dell'ambasciata.* [trad. 'In Switzerland, some militants climbed onto the roof of the embassy.']

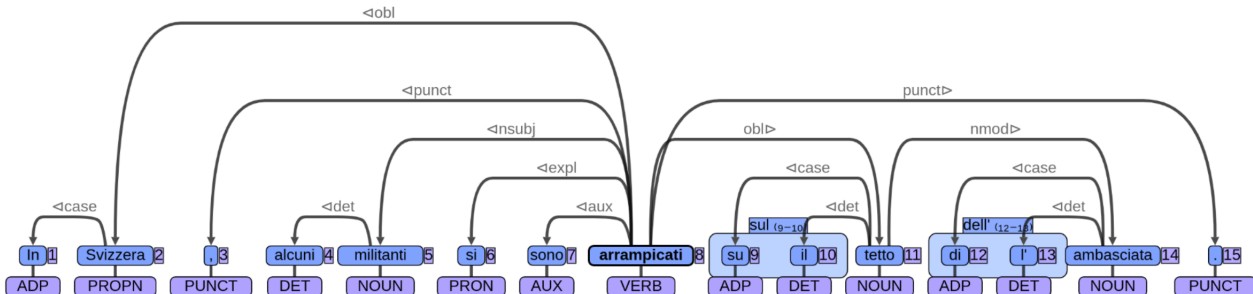

**Figure 1.** Linguistic annotation based on the UD scheme of the example sentence.

Relying on the morpho-syntactic level of IUDT annotation, we can observe, for example, that the above sentence features 20% of prepositions (ADP), 6.66% of verbs (VERB), and 20% of nouns (NOUN) out of the total number of parts of speech. Considering the features referring to the global syntactic structure, the depth of the whole syntactic tree of the sentence is equal to 3, corresponding to the two intermediate dependency links that are crossed in the path going from the root of the sentence (*arrampicati*, 'climbed') to each of the more distant leaf nodes, represented by the words *di* ('of') and *l'* ('the'), which compose the articulated preposition *dell'* dependent on the word *ambasciata* ('embassy'). Focusing on the local tree structure, the longest dependency relation is 6 tokens long, which corresponds to the number of tokens occurring linearly between the syntactic head *arrampicati* ('climbed') and the oblique object (obl) *Svizzera* ('Switzerland'), and we can observe a one-link-long prepositional complement chain (nmod) *dell'ambasciata* ('of the embassy') headed by the noun *tetto* ('roof'). In addition, the sentence is characterized by a canonical order of nuclear elements, since the nominal subject *militanti* ('militants') is in a pre-verbal position, which is the preferred order in Italian.

In this study, the values of each feature acquired from the IUDT represent the *gold values*. Table 2 reports the average distribution (*Mean*) and coefficient of variation (*CV*) of each group of linguistic features, computed as a mean of the values of every single feature included in the group. As can be noted, the mean values varied consistently across the three IUDT subsets, since we accounted for many different linguistic phenomena characterized by diverse ranges of values. As expected, most features were influenced by the length of the sentences being considered. In fact, while the mean values increased with the sentence length, the coefficients of variation, which captured the extent of values' variabilities within the same subset, tended to decrease as we approached the Longest subset. This suggests that, as sentences get longer, linguistic features tend to show higher but more stable values, while the opposite happened on sentences belonging to the Shortest subset. For the purposes of our experiments, the *gold values* reported in the gold dataset (IUDT) were automatically altered to generate *control datasets*.

**Table 2.** Average values and coefficients of variation of each macro-group of *gold* linguistic features extracted from the sentences in the Shortest, Standard, and Longest subsets of the IUDT.

| Feat. Group | Shortest | | Standard | | Longest | |
|---|---|---|---|---|---|---|
| | Mean | CV | Mean | CV | Mean | CV |
| Order | 19.83 | 1.04 | 40.45 | 0.55 | 52.96 | 0.34 |
| POS | 3.56 | 0.14 | 3.56 | 0.09 | 3.68 | 0.03 |
| Subord | 16.73 | 0.96 | 36.51 | 0.58 | 48.67 | 0.29 |
| SyntacticDep | 5.36 | 0.19 | 5.33 | 0.13 | 5.51 | 0.08 |
| TreeStructure | 4.62 | 1.17 | 11.66 | 0.56 | 17.37 | 0.29 |
| VerbInflection | 23.21 | 0.80 | 38.38 | 0.47 | 47.38 | 0.33 |
| VerbPredicate | 16.60 | 0.78 | 23.17 | 0.39 | 25.97 | 0.22 |

*3.3. Control Datasets*

We created two main types of control datasets for each subset of the IUDT, which were obtained by automatically altering the gold feature values according to different strategies. The first main type, hereafter referred to as *Swapped*, was built by shuffling the original values of each feature across sentences; the second type, *Random*, contained values that were randomly generated within the maximum and the minimum values that each feature showed in the gold datasets. To clarify, consider the following example involving the feature `average link length`, which captures the average linear distance between dependents and their syntactic head within a sentence. In the *Swapped* variant, we simply exchanged the feature values between sentences; thus, a sentence of the Standard subset that originally showed an `average link length` of, e.g., 2.86, could be changed to 8.83, a value that was originally associated with a different sentence. Note, in fact, that both are real values extracted from our dataset with respect to the considered feature, and they were simply randomly reassigned to a different sentence. When building the *Random* variant, all of the sentences here considered were associated with a feature value that was randomly generated between 1.33 and 9.78, and they were the reported minimum and maximum `average link length` values in the dataset (Standard subset).

Since the values of many considered features were highly influenced by the lengths of the sentences, we defined two additional alteration strategies to be combined with the main ones that accounted for such a property. In the first sub-type, *Bins*, we grouped sentences falling into the same predefined range of sentence lengths (i.e., 10–15, 15–20, 20–25, and 25–30 tokens). In a second sub-type, *Lengths*, we created groups of sentences with exactly the same length. Note that we applied these strategies only to sentences from the Standard subset, since the other two subsets did not present a considerable number of sentences for a given length.

Note that the different data-altering strategies were conceived to represent challenging testbeds to assess the effectiveness of our probing tasks in different scenarios. The *Swapped* control datasets were possibly the most challenging ones, as the swapped feature values might be quite similar to the gold ones and, thus, were possibly predicted with a high accuracy by the probing model. This intuition seemed to be confirmed by the differences between the values of the gold and each control dataset, which were obtained by averaging the differences between the gold and the altered values that each sentence had in the corresponding dataset. This held both in the Standard (Table 3) and in the Shortest and Longest subsets (Table 4). As can be noted, lower differences were reported for the Swapped control datasets, both on average and for each features group, in all subsets. Indeed, while the Random strategy tended to produce datasets where all possible values ranging between the maximum and minimum of that feature were equally distributed along sentences, the Swapped option simply shuffled gold values across sentences (namely, the mean value of a feature in the dataset did not change), producing untruthful but more plausible datasets.

**Table 3.** Average differences between the values of linguistic features in the *Gold* dataset and each *Control* dataset for each of the seven macro-groups.

| Group | Random | | | Swapped | | |
| --- | --- | --- | --- | --- | --- | --- |
| | Random | Bins | Lengths | Swapped | Bins | Lengths |
| Order | 0.48 | 0.48 | 0.48 | 0.41 | 0.40 | 0.40 |
| POS | 0.40 | 0.31 | 0.25 | 0.12 | 0.12 | 0.12 |
| Subord | 0.43 | 0.41 | 0.41 | 0.38 | 0.35 | 0.35 |
| SyntacticDep | 0.40 | 0.31 | 0.25 | 0.15 | 0.13 | 0.12 |
| TreeStructure | 0.36 | 0.28 | 0.25 | 0.20 | 0.18 | 0.18 |
| VerbInflection | 0.47 | 0.47 | 0.47 | 0.44 | 0.43 | 0.44 |
| VerbPredicate | 0.42 | 0.41 | 0.40 | 0.26 | 0.25 | 0.25 |
| Average | 0.42 | 0.38 | 0.36 | 0.28 | 0.27 | 0.26 |

**Table 4.** Average differences between the values of linguistic features in the *Gold* dataset and each *Control* dataset for each of the seven macro-groups considering only the Shortest and Longest subsets.

| | Shortest | | Longest | |
|---|---|---|---|---|
| **Group** | **Random** | **Swapped** | **Random** | **Swapped** |
| Order | 0.50 | 0.32 | 0.46 | 0.35 |
| POS | 0.43 | 0.13 | 0.38 | 0.13 |
| Subord | 0.49 | 0.20 | 0.39 | 0.28 |
| SyntacticDep | 0.44 | 0.13 | 0.37 | 0.15 |
| TreeStructure | 0.37 | 0.22 | 0.37 | 0.14 |
| VerbInflection | 0.50 | 0.34 | 0.44 | 0.42 |
| VerbPredicate | 0.46 | 0.24 | 0.39 | 0.21 |
| Average | 0.46 | 0.23 | 0.40 | 0.24 |

*3.4. Models*

For all experiments, we relied on a pre-trained Italian version of the BERT model, one of the most prominent NLMs. Specifically, we used the base-case BERT developed by the MDZ Digital Library Team, which was available through Huggingface's *Transformers* library [48] (https://huggingface.co/dbmdz/bert-base-italian-xxl-cased (accessed on 20 February 2023)). The model was trained by using the Italian Wikipedia and the OPUS corpus [49]. To obtain the sentence-level representations for each of the 12 layers of BERT, we leveraged the activation of the first input token *[CLS]*.

The probing model was a linear support vector regression model (LinearSVR). The model took as input the above layer-wise sentence-level representations, and it predicted the value of each considered feature in the Gold and Control datasets. Specifically, we trained and tested the probing model by adopting a cross-validation process on each dataset individually. To this aim, we split each dataset into five portions containing the same number of randomly selected sentences; then, we iteratively trained the probing model on four portions and used the remaining fifth as a test set. This way, the model was trained by using a representative sample of the dataset at each iteration.

As an evaluation metric, we used the Spearman correlation coefficient between the values of the linguistic features in the gold and control datasets and their values when predicted by the probing model by using BERT's sentence-level representations as input. In the remainder of this paper, we refer to the evaluation metric as the *probing score*.

Since previous work already showed the ability of pre-trained NLMs to outperform simple baselines (e.g., a linear model trained using only sentence length as an input feature) in the resolution of probing tasks [50], in this current paper, we did not perform a direct comparison with a baseline. Nevertheless, since the focus of this work is on assessing the sensitivity of BERT to distorted feature values, the control datasets can be viewed as baselines themselves.

**4. Results**

Our first analysis was devoted to assessing BERT's abilities in the prediction of the authentic values of the Gold dataset. These results represent the reference performance against which we compared the performance obtained on the diverse control datasets that we built. To better appreciate the impact of sentence length as a possible confounder of the probing approach that we devised, we kept separated the discussion of the results obtained on the Standard subset from the outcomes of the probing tasks performed on the Shortest and Longest subsets.

*4.1. Probing on the Standard Subset*

As a first analysis, we probed BERT's linguistic competence with respect to the seven groups of probing features. Figure 2 shows how the model's abilities to predict the considered linguistic phenomena in the Gold dataset changed across layers. As can be noted,

regardless of the group, BERT tended to lose knowledge as the output layer approached. As suggested by Liu et al. [31], this could be due to the fact that the representations that were better suited for language modeling were also those that exhibited worse probing task performance, indicating that the Transformer layers traded off between encoding general and probed features. However, in line with what was observed by Miaschi et al. [22,38] for the Italian and English languages, respectively, each group of features had a different behavior. Namely, the distributions of parts of speech (*POSs*) and dependency relations (*SyntacticDep*) were the best-encoded types of information, especially in the first layers; then, they constantly decreased. On the contrary, more complex linguistic knowledge about the order of subjects and objects with respect to the verb (*Order*) was acquired only in the middle layers. Notably, the model showed very scarce competencies concerning the number of dependents of a verbal head (*VerbPredicate*), which was quite constant across layers.

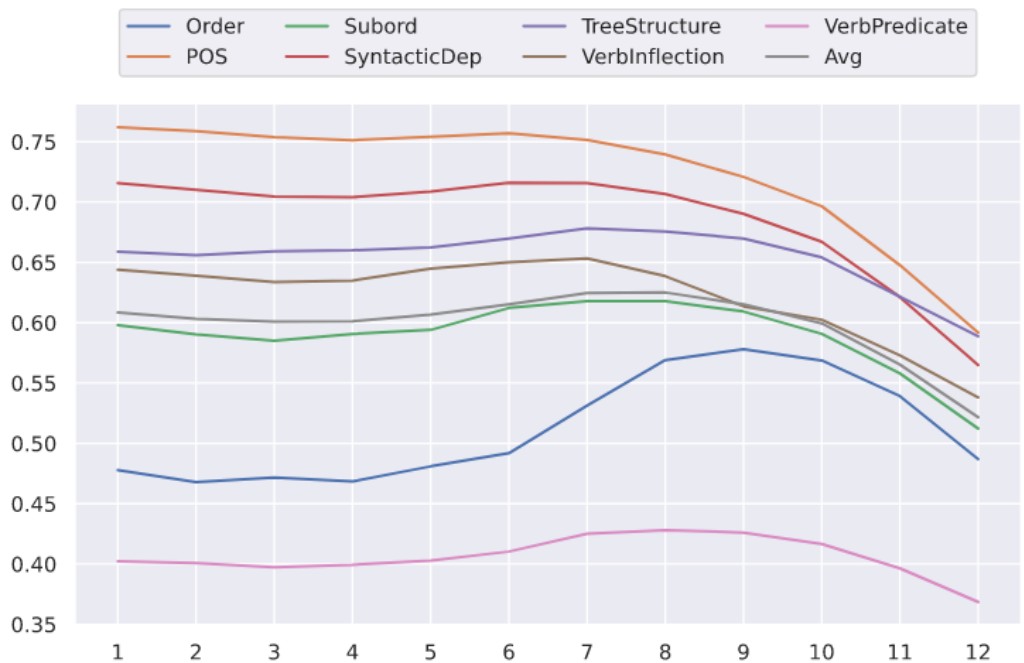

**Figure 2.** Layer-wise probing scores (Spearman correlations) obtained when predicting the *Gold* feature values of the Standard subset according to the seven macro-groups of linguistic features. Average results (*Avg*) are also reported.

To further investigate these trends across layers, for each feature, we computed the slopes of a linear regression line between BERT's layers and the values of the probing scores in the last and first layers. The *Gold* column of Figure 3 reports the slopes for the seven groups of features and for the total number of 77 features (line *Avg*). As can be noted, all of the slope values were negative, thus indicating that the learning curve decreased across layers. The only exception was represented by the trend of the features of the *Order* group, which had a positive value. this follows from the quite unique trend observed in Figure 2; the knowledge about this type of linguistic phenomenon, albeit very low, started to increase in the middle layers, and it decreased in the last ones, even though it remained higher with respect to the first ones. The features that BERT tended to know quite constantly across layers were those belonging to the *VerbPredicate* group. Accordingly, the slope value was the lowest one (−0.017).

Figure 3 also allows a first comparison between the performances of the probing model tested on the gold and control datasets. The majority of negative slope values reported here show that BERT's knowledge also generally tended to decrease across layers when tested against the different typologies of control datasets (Refer to Figure A1 for the layer-wise probing scores obtained on each control dataset). A few exceptions were

unevenly scattered across layers and groups of features, and they are not worth discussing. However, the most striking result emerging from Figure 3 was that the slopes were quite flat, both on average and considering specific features. Contrary to what was seen for the Gold dataset, we observed very small differences between the probing scores achieved by using the representations extracted from the last and first layers, indicating that the knowledge about linguistic features on all control datasets was stable across layers. This result seemed to suggest that altering the values of the gold features had a generic impact on BERT's linguistic knowledge.

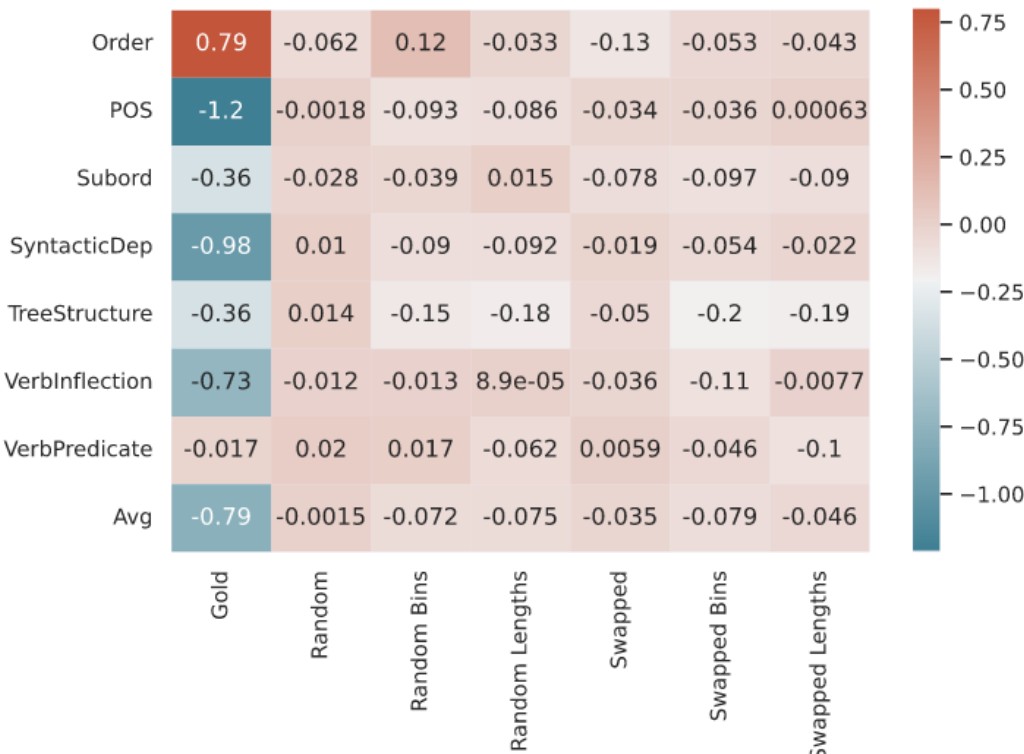

**Figure 3.** Slopes of the regression lines across the 12 layers for the probing scores obtained with the *Gold* and the corresponding *Control* datasets. Scores are multiplied by 100.

The extent of such an impact is clear by inspecting Figure 4, which reports the gaps between the probing scores obtained when predicting the gold and altered linguistic features. Here, we focused on the scores achieved in the output layer, since we previously observed very small changes in probing performance across layers. Specifically, the gap was computed as the difference between the probing score obtained at layer 12 on the Gold dataset and on each control dataset. Note that in order to weigh the impacts of the altered feature values with respect to BERT's competence for a given linguistic phenomenon, we divided the computed difference by the probing score obtained for each feature at layer 12 in the Gold dataset (The formula adopted for every single feature is the following one: (probing score at layer 12 in the Gold dataset—probing score at layer 12 in the control dataset)/probing score at layer 12 in the Gold dataset). Differences higher than 1 were obtained when the probing scores achieved on the control dataset were lower than 0.

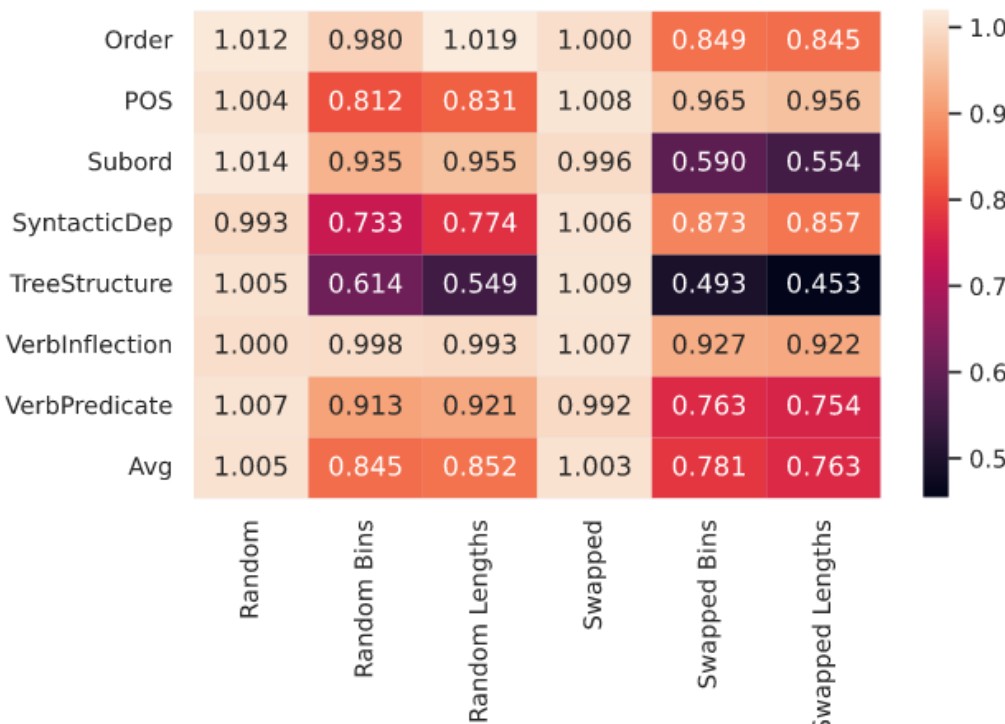

**Figure 4.** Differences between the probing scores obtained with the *Gold* dataset and each *Control* dataset by using the BERT representations extracted from the output (12) layer.

The positive value of differences visualized in the heatmap shows that, on average (*Avg* row), and for all groups of features, the highest probing scores were obtained on the Gold dataset even with some differences across the typologies of features and control datasets. The greatest differences were obtained for the *Random* and *Swapped* datasets without any constraints on the length of sentences. This seems to suggest that the probing model was able to recognize that the feature values contained in the two main types of control datasets were altered, even when they were not fully random, but plausible, i.e., in the *Swapped* datasets. As a consequence, we could hypothesize that the probing model was relying on some implicit linguistic knowledge when it predicted the authentic feature values, rather than learning some regularities that were possibly found in the dataset.

However, if we take a closer look at the gaps between the Gold and the altered datasets when we constrained the length of the sentences, we can observe that, on average (*Avg* row), the differences with respect to the prediction of the authentic feature values were generally lower. More specifically, the *Swapped Bins* (diff = 0.781) and *Lengths* (diff = 0.763) datasets were more challenging for our probing approach than the corresponding *Random* ones, against which we obtained higher differences equal to 0.845 and 0.852, respectively. Namely, since the feature values that were artificially created simply by shuffling gold ones across sentences constrained by sentence length were more similar to the gold values, as shown in Table 3, the swapped values were more confounding for the probing model. In fact, they were predicted with higher accuracy than randomly altered values.

In addition, stronger differences across groups of features emerged from this analysis. BERT's generalization abilities for features referring to the local and global syntactic structure of a sentence (*TreeStructure*) seemed the most similar to the gold ones based on the relatively small gap between predictions. Note that these sentence properties were the most sensitive to the sentence length, which BERT encoded with very high accuracy [51]. This may suggest that in the resolution of these tasks, the probing model possibly relied on some regularities related to sentence length. The same held for features related to *Subordination*, which were similarly highly correlated with sentence length. On the contrary, in both the *Swapped* and *Random* control datasets, the probing model performances diverged with

respect to the prediction of the pre- or post-verbal order of subject and object (*Order*) in a sentence and, in particular, of the verbal morphology features (*VerbInflection*), as shown by their smaller gaps.

### 4.2. Probing on the Shortest and Longest Subsets

In this section, we take a closer look at how BERT performed when tested against the *Shortest* and *Longest* subsets of IUDT sentences, which, as described in Section 3.1, gathered all sentences with a length of up to nine tokens and between 31 and 100 tokens, respectively. As in the previous section, we start by reporting the layer-wise probing scores obtained by the model when predicting the gold values of the linguistic features extracted from sentences belonging to these two subsets. These are shown in Figure 5, where we can see how BERT's implicit knowledge changed across layers and groups of linguistic phenomena. A first observation that we can draw from the figure is that the subset of long sentences exhibited a higher variation across layers, and this trend was more similar to the one observed for the *Standard* subset (see Figure 2). This held especially for some groups of phenomena, such as the distributions of parts of speech (*POS*s) and of dependency relations (*SyntacticDep*), for which BERT's predictions were very similar to the gold values, especially in the first layers, whereas this specific knowledge tended to decrease as the output layer was approached. A further similarity can be observed with respect to the worst encoded features, which were represented by sentence properties related to the complexity of verbal predicates (*VerbPredicate*) and, although to a lesser extent, to syntactic ordering (*Order*). Note that the latter group, as already observed for the *Standard* subset, was better encoded in the middle layers than in the first ones.

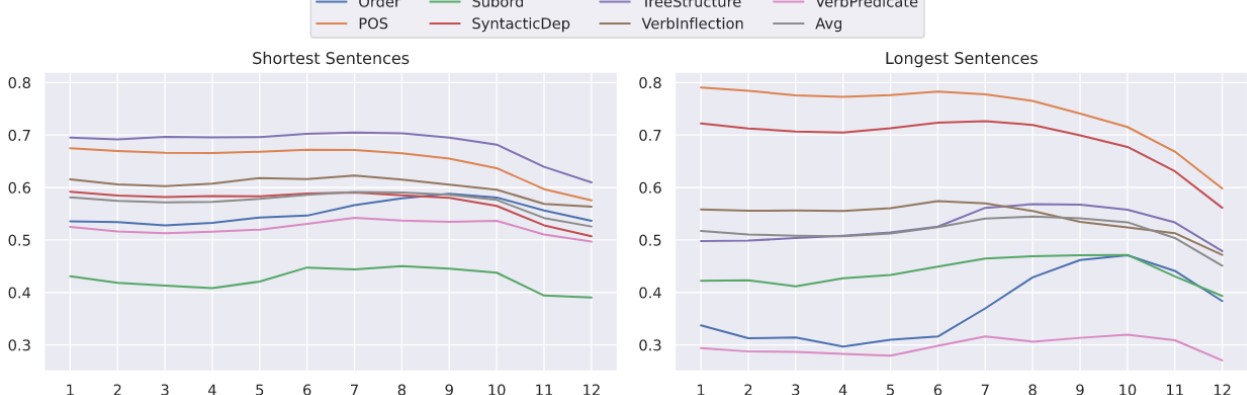

**Figure 5.** Layer-wise probing scores (Spearman correlations) obtained when predicting *Gold* feature values according to the seven macro-groups of linguistic features for the *Shortest* and *Longest* subsets. Average results (*Avg*) are also reported.

On the contrary, BERT's linguistic knowledge when tested on the subset of short sentences was, on average, more stable, with few variations across layers and across the diverse groups of linguistic phenomena. In addition, we can observe that BERT's competencies were differently ranked with respect to the ranking obtained in the *Standard* and *Longest* subsets. In fact, the features that the language model mastered with the highest accuracy were those modeling the syntactic structure of the sentence (*TreeStructure*), with a layer-wise average probing score equal to 0.68. Note that this score was higher than the accuracy achieved in the *Longest* (0.53) and *Standard* (0.65) subsets. This result may be a consequence of the fact that the values of the features belonging to this group were highly sensitive to sentence length, and short sentences were typically characterized by quite flat and simple syntactic trees (as shown in Table 2). Since, as we mentioned, sentence length was a feature that BERT mastered very well, BERT may have relied on the knowledge of this shallow feature as a proxy to predict more complex features related to the structure of the syntactic tree. Possibly related to the same reason, it turned out that BERT mastered

the order of subject and object (*Order*) and the number of dependents of verbal heads (*VerbPredicate*) much better in short than in long sentences, with accuracies even higher than those achieved on the Standard subset (The layer-wise average probing scores of the *Order* group were 0.55 in the *Shortest* subset, 0.37 in the *Longest* subset, and 0.51 in the *Standard* subset. The scores achieved for the *VerbPredicate* group were 0.52, 0.29, and 0.40 in the three datasets, respectively). Differently from the other two datasets, the worst prediction was achieved by the features modeling the subordination (*Subord*), even though they had probing scores very similar to those of the *Longest* subset.

Despite these differences, Figure 6 shows that BERT's knowledge tended to change very little across layers with respect to what was observed for the sentences in the *Standard* subset (Refer to Figure A2 for the layer-wise probing scores obtained on each control dataset). As previously noted, the average flattest slopes were obtained considering the Shortest subset (*Avg* = −0.49), while more variations could be seen for the Longest one. It is also worth highlighting that in the latter case, we had several groups of features with positive slope values. This was the case not only of the features belonging to the *Order* group, which had the same trend in the *Standard* and *Shortest* subsets, but also of the features modeling the subordination, the syntactic structure of a sentence, and the verbal arity.

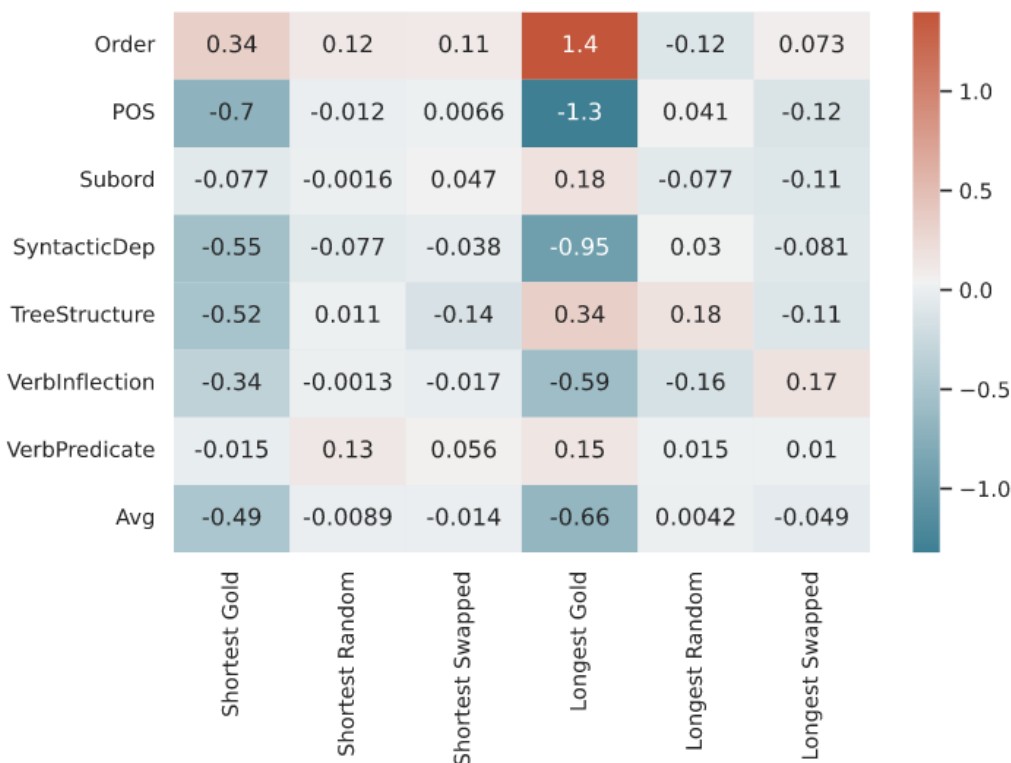

**Figure 6.** Slopes of the regression lines across the 12 layers for the probing scores obtained with the *Gold* and the corresponding *Control* datasets for the Shortest and Longest subsets. Scores are multiplied by 100.

In addition, the figure allows a first analysis of the impact of the corresponding control datasets on the probing model's performance. Specifically, we can see that the altered feature values were predicted quite similarly across layers, while the prediction of the gold values underwent more variations. This trend was similar to the one reported for the Standard subsets, and it suggests that, in less standard sentences, the probing model is also sensitive to the distortion of feature values.

Further evidence in this direction can be acquired by inspecting Figure 7, which reports the gap between the probing model's accuracy on the *Gold* and *Control* datasets for the two

subsets. As noted in the previous section, the positive values show that the gold values of the features were predicted with higher accuracies than the altered ones (In addition, in this case, the differences were weighted based on the probing scores obtained by each feature on the gold *Shortest* and *Longest* subsets). As in the case of the *Standard* subset (see Figure 4), very few variations across the groups of features emerged, thus showing that the probing model was scarcely confused by the distortion of feature values, regardless of the linguistic phenomenon tested. We noticed, for example, that BERT's knowledge concerning the use of subordination was lower both in the *Shortest* and *Longest* subsets than in the *Standard* one. However, the gap between the probing scores obtained for the corresponding *Gold* and *Control* datasets was similarly high in the three subsets. However, differently from what we observed for the *Standard* subset, the *Swapped* control datasets were slightly more challenging than the *Random* ones. In fact, the differences were, on average (*Avg* row), lower, especially when the probing model was tested against the control datasets of the short sentences.

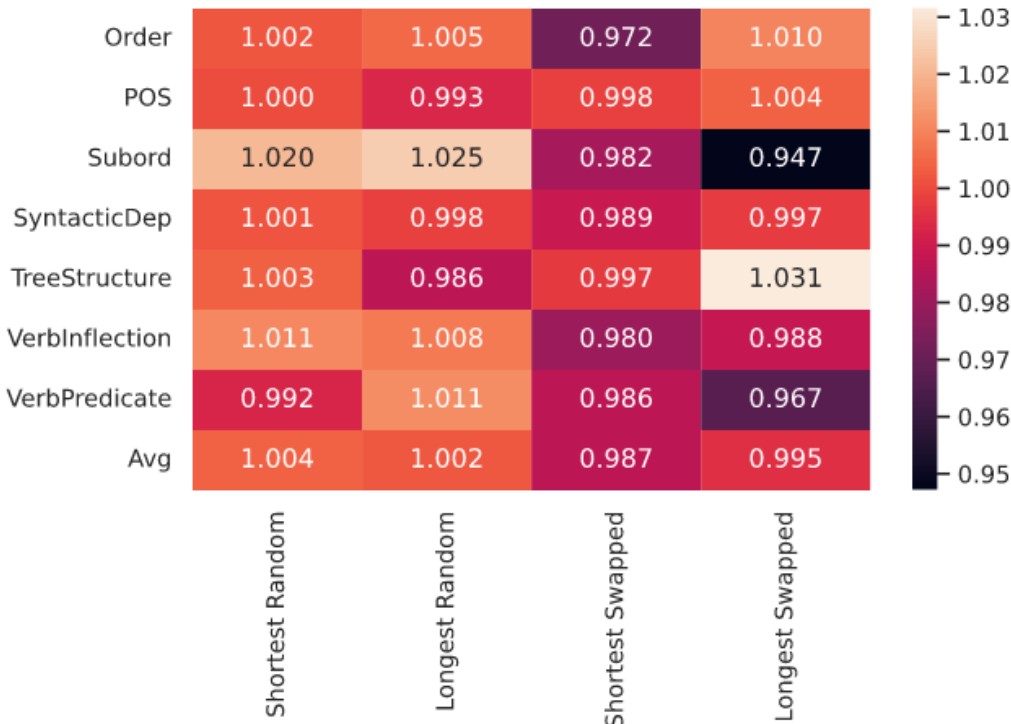

**Figure 7.** Differences between the probing scores obtained for the *Gold* dataset and each *Control* dataset for the *Longest* and *Shortest* subsets. Differences were computed by using BERT representations extracted from the output (12) layer.

## 5. Conclusions

In spite of the large number of studies that have relied on the diagnostic probing paradigm to assess the linguistic knowledge implicitly encoded by an NLM's representations, the validity of this method is still questionable from different perspectives. Our study has presented a novel contribution to this debate by focusing specifically on one of the still-open questions, that is, the effectiveness of probes in reflecting the linguistic properties encoded in a representation. To this aim, we analyzed the performance of a probing model trained with layer-wise sentence-level BERT representations to predict the value of a large set of linguistic features derived from the Italian Universal Dependency Treebank (IUDT) and from a suite of control datasets that were specifically created to alter the original values of the examined features.

As a general remark, we observed that the probing model always had a better performance when tested against the IUDT datasets than it did when tested against the

corresponding control datasets. Namely, the gold values of the considered set of linguistic features were predicted with higher accuracy than with the artificially altered ones, thus showing that the probing model was sensitive to the distortion of feature values, and it does not simply learn the regularities of the probing task. This result corroborates the reliability of the probing task as an interpretability approach for assessing the level of linguistic knowledge implicitly encoded in BERT's sentence-level representations.

However, our experiments also highlighted that sentence length is a relevant confounding factor that may bias the real estimate of BERT's linguistic knowledge. In fact, when we focused on sentences of the same length (or in the same ranges of lengths) taken from the *Standard* IUDT subset, we observed that the probing model was less sensitive to the artificially generated values of features, especially when these values were obtained by shuffling the original values across sentences of the same length (or ranges of lengths). This suggests that, when the length is controlled, an alteration strategy that assigns incorrect but still plausible values is more challenging for the probing model than one that simply generates random values. This general trend concerns, in particular, groups of linguistic phenomena that are more influenced by the length of a sentence. This is the case, for example, of features that model local and global characteristics of the syntactic structure of a sentence (i.e., the *TreeStructure* group), which tend to have quite homogeneous values within sentences of the same length. Accordingly, the output space of the probing model for these features is smaller than in the whole dataset, thus making them more easily predictable without relying on authentic linguistic competence. Despite this trend being particularly visible when we consider the output layer, we showed that the probing model was also sensitive to the altered values across BERT's twelve layers. Quite interestingly, contrary to what was observed for the Gold dataset, the learning curve of the model tested on the control datasets decreased quite slowly across layers, with no significant variations across the typologies of linguistic phenomena.

The main outcomes obtained for the group of sentences with a standard length in Italian were also confirmed by the experiments conducted on the subsets of sentences with less standard lengths. Although we highlighted that BERT mastered specific linguistic phenomena with different accuracies in the *Shortest*, *Longest* and *Standard* subsets, we showed that the probing model was similarly scarcely confused in the three subsets, regardless of the linguistic aspect considered. This seems to suggest that BERT's representations extracted from less standard sentences implicitly encoded the linguistic knowledge of the phenomena therein.

The present study can be extended from various perspectives. In the future, the effectiveness of the diagnostic probing approach can be evaluated by considering other languages—possibly those belonging to different language families and, thus, characterized by different feature values. Indeed, it could be worth exploring whether confounding factors affecting the performance of probing models are shared among languages or vary depending on their family. In this respect, we can either reuse the same set of linguistic features or focus on subsets of phenomena of particular interest for a typological study. In fact, the approach adopted to select the set of linguistic features is multilingual, as it is based on the Universal Dependencies formalism. In addition, as neural models continue to improve, a further possible direction of research may consist of assessing the effectiveness of the probing approach in testing the linguistic knowledge encoded in models with different architectures.

**Author Contributions:** Conceptualization, methodology, validation, resources, A.M., C.A., D.B., F.D. and G.V.; formal analysis, investigation, A.M. and F.D.; data curation, C.A., D.B. and G.V.; writing—original draft preparation, A.M., C.A., D.B. and G.V.; writing—review and editing, A.M., C.A., D.B., F.D. and G.V.; visualization, A.M. and C.A.; supervision, project administration, F.D. All authors have read and agreed to the published version of the manuscript.

**Funding:** This research received no external funding.

**Institutional Review Board Statement:** Not applicable.

**Informed Consent Statement:** Not applicable.

**Data Availability Statement:** The Italian treebanks used in the study are publicly available at the following link: https://lindat.mff.cuni.cz/repository/xmlui/handle/11234/1-3105 (accessed on 20 September 2022). The gold/control datasets and the results obtained by the probing model trained with BERT's internal representations: https://github.com/alemiaschi/Testing_the_Effectiveness_of _the_Diagnostic_Probing_Paradigm_Supplementary_Materials (accessed on 20 September 2022).

**Acknowledgments:** We acknowledge the project "Human in Neural Language Models" (*IsC93_HiNLM*), funded by the CINECA (https://www.cineca.it/en (accessed on 20 September 2022)) under the ISCRA initiative, for the availability of HPC resources and support.

**Conflicts of Interest:** The authors declare no conflict of interest.

**Appendix A**

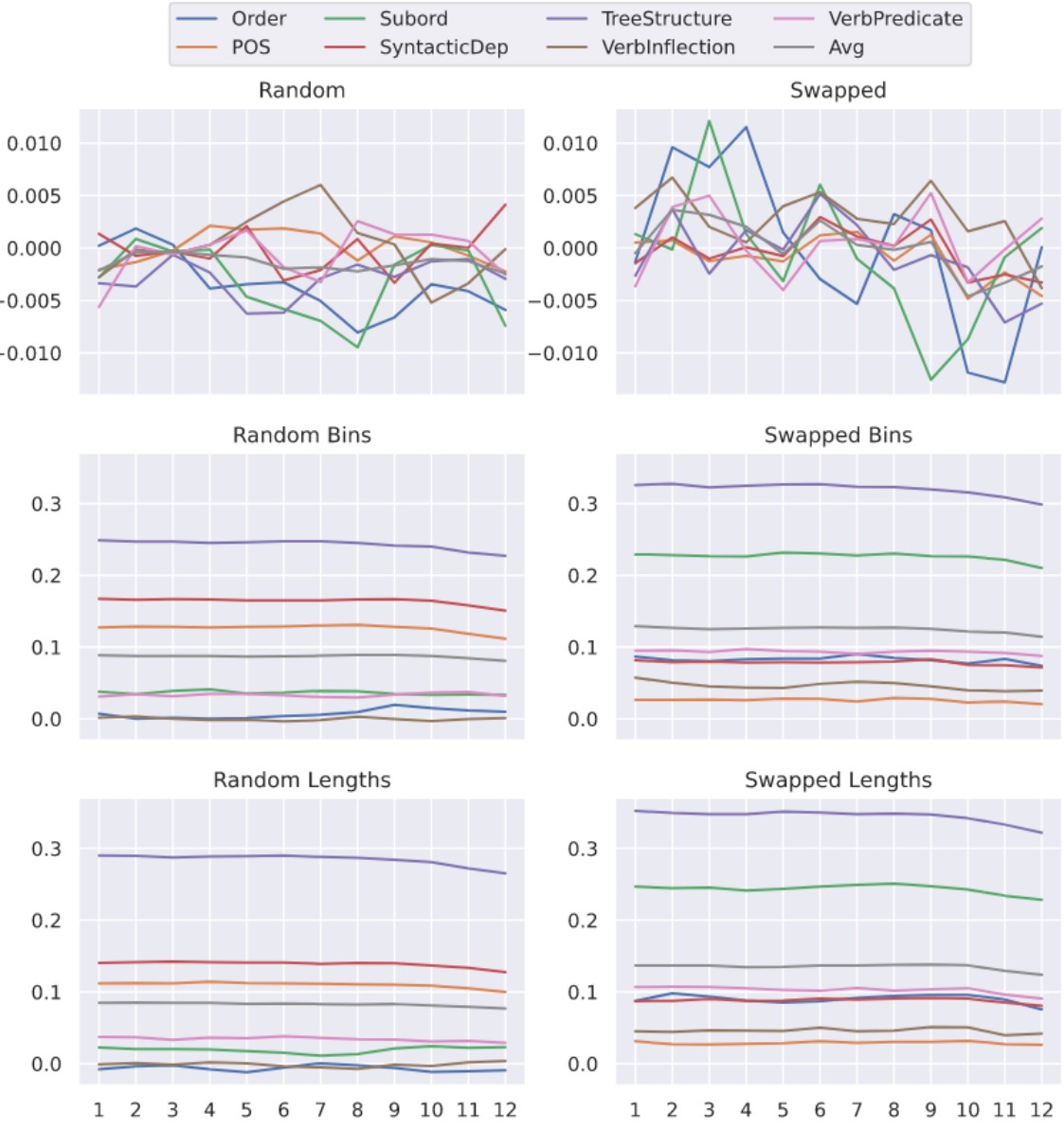

**Figure A1.** Layer-wise probing scores (Spearman correlations) obtained when predicting *Control* feature values of the Standard subset according to the seven macro-groups of linguistic features. Average results (*Avg*) are also reported.

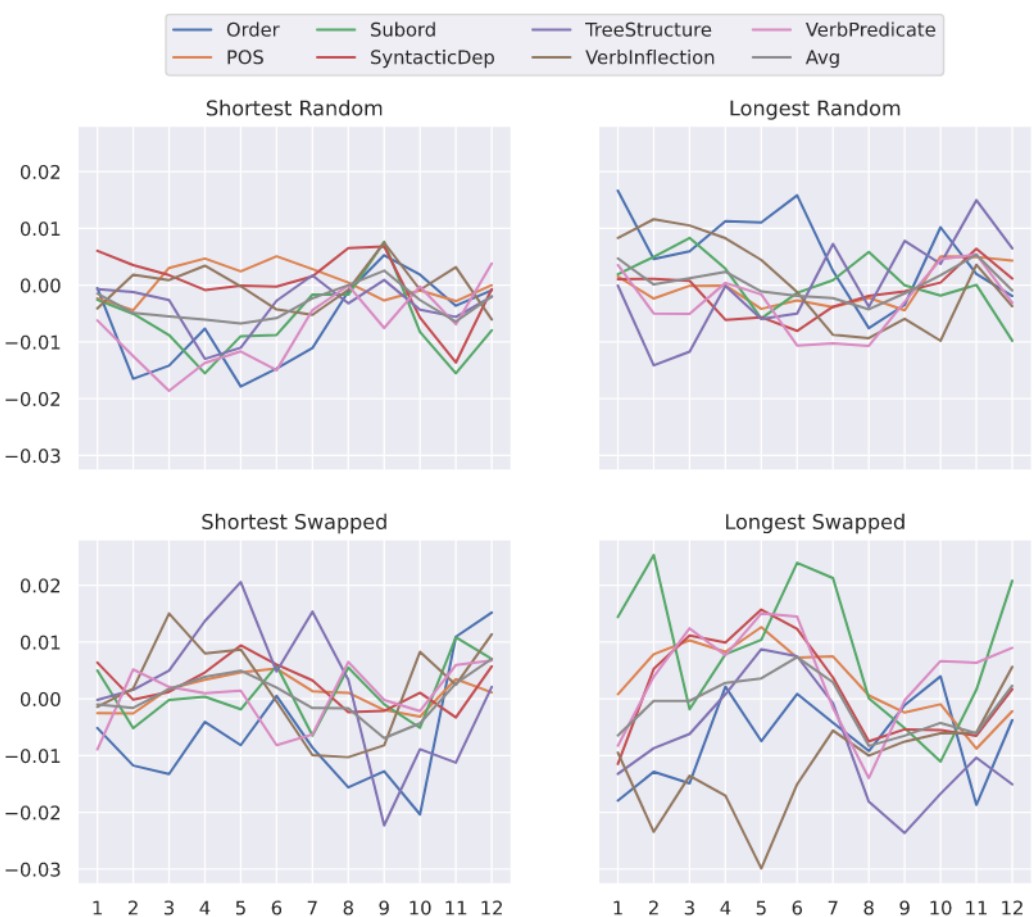

**Figure A2.** Layer-wise probing scores (Spearman correlations) obtained when predicting *Control* feature values according to the seven macro-groups of linguistic features for the *Shortest* and *Longest* subsets. Average results (*Avg*) are also reported.

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
