# Peer review of "Testing the Effectiveness of the Diagnostic Probing Paradigm on Italian Treebanks"

_information, doi:10.3390/info14030144_

Round 1

Reviewer 1 Report

This is a very good paper questioning the effectiveness of the probing task paradigm in analyzing the linguistic competence of neural language models: it does so by comparing the performance of a BERT vectors-based regressor on a dataset with gold linguistic features extracted from an italian treebank, and modified versions of the same dataset (the so-called "control datasets") where the target values have been altered according to different strategies. Additionally, the authors control for the length of the sentences, in order to check the influence of this parameter on the model prediction. The results show that the model performance is significantly lower in the control datasets, suggesting that the successful BERT predictions for the actual gold standard value are actually relying on linguistic knowledge absorbed by the language model. Moreover, sentence length has an important role in the predictions, as applying the alteration strategies to group of sentences of controlled length leads to more plausible target values for the control datasets.

I only have a few questions for the authors:

- if I understand well, in the Swapped strategy for each instance in your dataset you just take another instance and swap the target value. Is it possible that a given instance is selected for swapping multiple times? If yes, as I assume that the feature values are more or less normally distributed, aren't you risking that mid-range values are overrepresented in the control dataset?

- related to the previous question, I think it would be interesting to split the target values of the control datasets into percentiles, and investigate the cumulative prediction error for percentile ranges. My intuition would be that, for very extreme values in the control datasets, the predictions should be much closer to the original gold values and further away from the replaced ones;

- a small clarification on the contribution: since the authors themselves present the paper as an extension of the methods proposed in the previous works by Miaschi et al., in what sense is this paper the first one "testing the effectiveness of diagnostic probing in Italian"? Weren't the previous papers on Italian as well or...? Or are you referring to the specific approach of introducing control datasets as a reliability check for the probing itself?

TYPOS/MINOR THINGS:

- l. 52 Starting and extending the methodology --> Starting with and extending the methodology

- l. 72 one of the most addressed lines of research --> Maybe better "one of the most popular" in this context?

SUGGESTED REFS:

- as for the explainability methods of Transformers' predictions (section 1), you might be interested in citing the recent works by Hollenstein and colleagues, e.g. [1] and [2].

[1] Hollenstein and Beinborn, 2021.  Relative Importance in Sentence Processing. Proceedings of ACL-IJCNLP.

[2] Morger et al., 2022. A Cross-lingual Comparison of Human and Model Relative Word Importance. Proceedings of the 2022 CLASP Conference on (Dis)embodiment.

Reviewer 2 Report

The paper proposes an interesting analysis of robustness testing and the effectiveness of language models for capturing linguistic features. The paper is well-structured and well-written. The introduction highlights and contextualizes simply the current problem, as well as the terms and concepts that make up the theoretical framework. The contributions are provided concretely and extensive and rich related works are presented, in which the authors do not lose sight of their work at all times, contextualizing and differentiating. The paper also highlights the need for language models and testing techniques in languages other than English, in this case, Italian. 

The methodology of the experimentation, presenting a correct battery of tests and test variations in two scenarios, one with the unmodified dataset and the other with slight variations, I conclude that it is a solid validation technique, although I would like to have seen the results with another language model or probing algorithm.  The results of the paper are well interpreted, and well presented and the conclusions are perfectly supported by these results. 

A concern with this paper is that some of the novelty of the paper is based on conference papers by the same authors of the paper, which are correctly cited and framed, but which undoubtedly reduces the novelty of the paper to a large extent. Even so, I consider that this is not a problem and that the paper can be published in the journal after considering reviewing the following: 

-On line 301, the authors indicate that they cannot compare the system with some baseline model. I think that at least a comparison could be added, or justification that the conclusions of the paper are supported not only by BERT. Consider adding another probing model to the experimentation, as the results would gain strength with this contrast.

-Consider adding a paragraph in the introduction where the structure of the paper is made clear.
